# A Multi-Omics Network of a Seven-Gene Prognostic Signature for Non-Small Cell Lung Cancer

**DOI:** 10.3390/ijms23010219

**Published:** 2021-12-25

**Authors:** Qing Ye, Brianne Falatovich, Salvi Singh, Alexey V. Ivanov, Timothy D. Eubank, Nancy Lan Guo

**Affiliations:** 1West Virginia University Cancer Institute, Morgantown, WV 26506, USA; qiye@mix.wvu.edu (Q.Y.); bf00014@mix.wvu.edu (B.F.); ss0083@mix.wvu.edu (S.S.); aivanov@hsc.wvu.edu (A.V.I.); tdeubank@hsc.wvu.edu (T.D.E.); 2Lane Department of Computer Science and Electrical Engineering, West Virginia University, Morgantown, WV 26506, USA; 3Department of Biochemistry, School of Medicine, West Virginia University, Morgantown, WV 26506, USA; 4Department of Microbiology, Immunology and Cell Biology, School of Medicine, West Virginia University, Morgantown, WV 26506, USA; 5Department of Occupational and Environmental Health Sciences, School of Public Health, West Virginia University, Morgantown, WV 26506, USA

**Keywords:** non-small cell lung cancer, prognostic gene signature, multi-omics, CRISPR-Cas9, RNA interference, Boolean implication networks, chemotherapy, immune checkpoint inhibitor, radiotherapy, drug reposition

## Abstract

There is an unmet clinical need to identify patients with early-stage non-small cell lung cancer (NSCLC) who are likely to develop recurrence and to predict their therapeutic responses. Our previous study developed a qRT-PCR-based seven-gene microfluidic assay to predict the recurrence risk and the clinical benefits of chemotherapy. This study showed it was feasible to apply this seven-gene panel in RNA sequencing profiles of The Cancer Genome Atlas (TCGA) NSCLC patients (*n* = 923) in randomly partitioned feasibility-training and validation sets (*p* < 0.05, Kaplan–Meier analysis). Using Boolean implication networks, DNA copy number variation-mediated transcriptional regulatory network of the seven-gene signature was identified in multiple NSCLC cohorts (*n* = 371). The multi-omics network genes, including *PD-L1*, were significantly correlated with immune infiltration and drug response to 10 commonly used drugs for treating NSCLC. ZNF71 protein expression was positively correlated with epithelial markers and was negatively correlated with mesenchymal markers in NSCLC cell lines in Western blots. PI3K was identified as a relevant pathway of proliferation networks involving *ZNF71* and its isoforms formulated with CRISPR-Cas9 and RNA interference (RNAi) profiles. Based on the gene expression of the multi-omics network, repositioning drugs were identified for NSCLC treatment.

## 1. Introduction

Lung cancer has the second highest cancer incidence rate and the top cancer-related mortality worldwide. An estimate from the American Cancer Society shows that, in 2021, there were 235,760 lung cancer cases (119,100 men and 116,660 women) in the US [1]. In the US, 60–65% of newly diagnosed lung cancer cases are never-smokers or former smokers [2]. About 84% of all known lung cancers cases are non-small-cell lung cancer (NSCLC) [3]. Major histological subtypes of NSCLC include lung adenocarcinoma (LUAD, 50% of NSCLC cases), squamous cell carcinoma (LUSC, 30%), and large cell carcinoma (5–10%) [4]. The overall 5-year survival rate after initial diagnosis for NSCLC is less than 15% due to extensive invasion, limited therapeutic response, recurrence, and metastasis [5]. Patients with NSCLCs that are in stage II and above are treated with chemotherapy. Patients in stages III and IV are treated with additional radiotherapy [6]. Although adjuvant chemotherapy for stage II and stage III disease improves overall survival rate by 10–15% [7], early-stage NSCLC has a dismal prognosis, with a 5-year mortality rate of 40% in stage I, 66% in stage II, and 75% in stage IIIA, as a result of recurrence [8]. These data suggest that many early-stage tumors harbor occult recurrence despite chemotherapy. The FDA recently approved atezolizumab for stage II and IIIA NSCLC after platinum-based chemotherapy in patients with PD-L1 > 1% [9]. There is an unmet clinical need to identify early-stage NSCLC patients who are likely to develop recurrence and to predict the clinical benefits of chemotherapy, with or without subsequent immunotherapy depending on their PD-L1 expression levels.

Our previous study developed a qRT-PCR-based seven-gene microfluidic assay for NSCLC prognosis and prediction of chemotherapeutic benefits, including *ABCC4*, *CCL19*, *SLC39A8*, *CD27*, *FUT7*, *DAG1*, and *ZNF71* [10]. This gene assay can inform selection of specific chemotherapy for individual patients to improve cancer care. Zinc finger protein 71 (ZNF71) protein expression quantified with AQUA was identified as a good prognostic marker of NSCLC [10]. As a transcriptional repression domain, the Krüppel-associated box (KRAB) is commonly found in human zinc finger protein-based transcription factors, specifically KRAB zinc finger proteins (KRAB-ZFPs) [11,12,13]. KRAB-ZFPs-mediated transcriptional repression is involved in cell proliferation, differentiation, apoptosis, and cancer [14]. Emerging evidence links epithelial to mesenchymal transition (EMT) to tumor progression, metastasis, and resistance to cancer therapy [15,16,17,18]. We reported that *ZNF71 KRAB* isoform was associated with EMT in NSCLC tumors and cell lines [19].

Revelation of genomic and transcriptomic landscape has shed lights to NSCLC subtypes and treatment [20]. An examination of TCGA multi-omics data showed that these seven prognostic genes have more DNA copy number variations (CNV) than mutations in NSCLC tumors (Figure A1, Figure A2, Figure A3, Figure A4, Figure A5, Figure A6 and Figure A7). Here, we sought to investigate 1) the feasibility of applying the seven-gene signature for NSCLC prognosis using next-generation sequencing (NGS) data of TCGA; and 2) the potential functional mechanisms involving the molecular networks of these seven genes. In order to understand how the molecular networks enfold these seven prognostic genes, we developed a novel Boolean implication network methodology for modeling whole-genome copy number variation (CNV) and gene expression profiles (*n* = 371) in NSCLC tumors. CRISPR-Cas9 (*n* = 78) and RNAi (*n* = 92) screening data were used to identify proliferation genes in the multi-omics network associated with the seven-gene panel. The identified network genes were further examined for their association with response to radiotherapy in TCGA patient tumors (*n* = 966) and 10 commonly used NSCLC therapeutic regimens in the Cancer Cell Line Encyclopedia (CCLE) NSCLC panel (*n* = 117). In myeloid dendritic cells, macrophages, neutrophils, CD4+ T cells, CD8+ T cells, and B cells, immunological infiltration linked to the discovered genes was studied. The association between mRNA expression and protein expression of these network genes in NSCLC CCLE panel and TCGA tumors was analyzed. 

EMT is highly dynamic and reversible. EMT is still unknown and controversial as to its role in cancer outcome [21,22]. There was a reported link between KRAB-ZFPs and EMT [23]. Tripartite motif containing 28 (TRIM28) protein is a universal co-factor for KRAB-ZFP transcription factors [24]. TRIM28 contributes to EMT and might be involved in lung cancer metastasis [25]. We previously reported that *ZNF71 KRAB* was associated with EMT and patient outcomes [19]. Nevertheless, the function of *ZNF71 KRAB* in EMT is not known. To better understand the molecular mechanisms of *ZNF71* and its isoforms, this study investigated the correlation between ZNF71 protein expression and EMT markers in NSCLC cell lines. Furthermore, molecular networks formulated with proliferation genes co-expressed with *ZNF71*, *ZNF71 KRAB*, and *ZNF71 KRAB*-less isoforms in NSCLC tumors were constructed with the Boolean implication network algorithm, respectively. The gene expression of these proliferation networks was further analyzed with L1000 Connectivity Map (CMap) [26,27] to identify matched functional pathways in shRNA knockdown, CRISPR-Cas9 knockout, or overexpression assays in NSCLC cell lines. Finally, the 14-gene EMT classifier constructed in our previous study for characterizing NSCLC tumor EMT states [19] was combined with the identified multi-omics network of the seven-gene signature to generate mechanistic hypotheses and discover repositioning drugs to reverse EMT, to enhance drug sensitivity, and to inhibit ICIs and NSCLC proliferation. Such a revelation will provide insight for designing therapeutic strategies for NSCLC intervention in future research.

## 2. Results

### 2.1. A Multi-Omics Network of the Seven-Gene NSCLC Prognostic and Predictive Signature

Our previous retrospective longitudinal clinical studies developed a seven-gene qRT-PCR microfluidic assay for NSCLC prognostic and prediction of chemotherapeutic benefits [10], including *ABCC4*, *CCL19*, *SLC39A8*, *CD27*, *FUT7*, *DAG1*, and *ZNF71*. Here, we sought to investigate whether it is feasible to apply this seven-gene signature in NGS data for NSCLC prognosis. TCGA-LUAD (*n* = 515) and TCGA-LUSC (*n* = 501) datasets were combined as TCGA NSCLC data (*n* = 923), keeping patient samples with sufficient survival information. The combined TCGA NSCLC dataset was randomly partitioned into a training (*n*= 462) and a testing set (*n*= 461) for building and validating the prognostic model, respectively. A multivariate Cox model was built using the seven prognostic genes in the training set to calculate coefficients for the gene covariates and the risk-score for each patient. This model was applied to the testing set using the same gene coefficients and cutoff for patient stratification. In Kaplan–Meier analysis, the patients who had a risk-score less than −0.74 survived significantly longer than the patients who had a risk-score equal or greater than −0.74 in both training and testing sets (*p* = 0.016, HR: 1.454 [1.072, 1.971] in training set, *p* = 0.049, HR: 1.366 [1.001, 1.864] in the testing set; Figure 1A,B) in Kaplan–Meier analysis. More details were included in Appendix A. 

The Boolean implication network approach was used to create a multi-omics association network of *ABCC4*, *CCL19*, *SLC39A8*, *CD27*, *FUT7*, *DAG1*, and *ZNF71*. The following genes were selected to build the network. First, the genome-scale analysis identified genes that had significant (*p* < 0.05, *z*-tests) co-expression with at least two of the seven original genes in both the GSE31800 and GSE28582 NSCLC patient cohorts. Second, genes with a significant (*p* < 0.05, *z*-tests) CNV-mediated gene expression (GE) associated with at least two of the seven original genes in both GSE31800 and GSE28582 cohorts were identified. Third, we screened for genes in both GSE31800 and GSE28582 that had a significant (*p* < 0.05, *z*-tests) co-occurrence of DNA copy number aberrations with at least two of the seven original genes, but no genes were discovered. In addition, proliferation genes were defined as genes with a significant dependency score in 50% of the tested NSCLC cell lines in CRISPR-Cas9 (*n* = 78) or RNAi (*n* = 92). *ERH*, *IFITM3*, *NFS1*, and *KRT17* were identified as NSCLC proliferation genes. The final multi-omics network (Figure 1C) contained genes having an association with at least two of the seven prognostic genes, with each edge indicating an observed co-expression or CNV-mediated gene expression dysregulation in NSCLC tumors in multiple patient cohorts. *PD-L1 (CD274*) was co-expressed with multiple signature genes. Down-regulation of *CD274* was associated with up-regulation of *DAG1*, and down-regulation of *DAG1* was associated with up-regulation of *CD274*. Up-regulation of *CD274* was associated with up-regulation of *CCL19* and down-regulation of *CD27* (Appendix A). It was reported that X-ray irradiation induced miR-574-3p expression in NSCLC A549 cells, which, in turn, suppressed enhancer of rudimentary homolog (ERH) protein production and delayed cell growth [28]. Interferon-induced transmembrane protein 3 (IFITM3) protein expression was expressed in the cytoplasm of metastatic lung adenocarcinoma tumor tissues and was correlated with higher tumor grade [29]. Single-cell sequencing profiles of *IFITM3* and *ISG15* characterize an interferon signaling cluster of macrophages in metastasis-bearing lungs [30]. *NFS1* cysteine desulfurase is under positive selection in lung adenocarcinomas and protects cancer cells from ferroptosis, a non-apoptotic form of cell death, in response to oxidative damage [31]. Keratin 17 (*KRT17*), reported to promote lung adenocarcinoma via enhancing cell proliferation and invasion [32], showed a paradoxical anti-metastatic role of *KRT17* both in vitro and in vivo in cancer xenografts in a single-cell lineage study [33]. These reports in the literature support the relevance of the identified multi-omics network of the seven-gene signature in NSCLC.

Stage III and IV NSCLC patients receive radiation therapy [6]. To identify genes associated with the response to radiotherapy, TCGA-LUAD and TCGA-LUSC stage III and IV patients who had received radiotherapy were included in the analysis. There was significant differential expression (*p* < 0.05, two sample *t*-tests) in genes identified in the multi-omics network (Figure 1C), including *SERPINB5, TTC16,* and *CD27*. These differences were observed in the short survival (<20 months) and long survival (>58 months) patient groups (Figure 1D). There was a higher level of expression of *CD27* and *TTC16* among the short-survival patients, and higher levels of *SERPINB5* expression among the long-survival patients. 

The correlation of the expression of 65 genes (*DEFB103A* was not available) from the final network with immune infiltration in TCGA-LUAD (*n* = 515) and TCGA-LUSC (*n* = 501) patients was assessed with TIMER 2.0 (Figure 2A,B) [34,35,36]. The immune infiltration analysis included myeloid dendritic cells, macrophages, neutrophils, CD4+ T cells, CD8+ T cells, and B cells. All the available genes in the 7-gene network had a significant correlation with immune infiltration in at least one examined cell types. 

This study also examined the association with drug activities of 10 commonly used therapeutic regimens for treating NSCLC, including carboplatin, cisplatin, paclitaxel (Taxol), pemetrexed, docetaxel, gemcitabine, vinorelbine, etoposide, gefitinib, and erlotinib. There were three histological subtypes for the CCLE NSCLC cell lines: adenocarcinoma, squamous cell carcinoma, large cell carcinoma, and adenosquamous carcinoma. Our study selected genes associated with drug response based on their differential expression of mRNA and protein in sensitive and resistant NSCLC cell lines. In the identified seven-gene network (Figure 1C), numerous genes were associated with drug sensitivity or resistance to the studied drugs in the CCLE NSCLC panel (Table 1 and Table 2).

Next, we examined the correlation between mRNA expression and protein expression of the seven-gene network in NSCLC cell lines and patient tumors. In CCLE NSCLC cell lines (*n* = 63), 41 genes in the seven-gene network had both mRNA and protein expression measurements. A total of 34 out of 41 genes had significant correlation between mRNA expression and protein expression in NSCLC cell lines (*p* < 0.05, Pearson’s correlation; Figure 3A). TRIM29 had two corresponding protein expression quantifications, both significantly correlated with its mRNA expression. The proportion of genes with a significant mRNA and protein expression correlation is 82.9% in the identified seven-gene network, which is significantly higher than 66.3% in the remaining 11,452 genes in the genome (two-tailed proportion *z*-test *p* < 0.05, Figure 3B). In TCGA-LUAD and TCGA-LUSC tumors, only CDH3 and CD274 had both mRNA expression and protein expression. The protein expression of CDH3 was positively correlated with its mRNA expression in TCGA NSCLC patient tumors (*p* < 0.05, Pearson’s correlation; Figure 3C), consistent with the observation in CCLE NSCLC cell lines (Figure 3A).

### 2.2. Gene Co-Expression Networks of ZNF71 and Isoforms

In the study cohorts, *ZNF71* was associated with a more favorable survival outcome among patients who received cisplatin, carboplatin, and Taxol, indicating its linkage with chemosensitivity [10]. While *ZNF71* mRNA expression was not associated with survival in NSCLC patient tumors analyzed in qRT-PCR, a higher ZNF71 protein expression measured with AQUA linked to a longer overall survival in two separate NSCLC cohorts (*n* = 291) using tissue microarrays [10]. The ZNF71 antibody (Abcam; ab87250) used in our previous study [10] was discontinued. Protein level of overexpressed ZNF71 (GeneTex, Cat. No. GTX116553) was detected in our previous study [19]. 

Gene *ZNF71* gene is composed four exons, in which exons 1 and 2 encode the 180nt 5′UTR and the first 11 amino acids of the protein. The next 43 amino acids (aa) in exon 3 are the majority of the KRAB repression domain. As the longest exon, exon 4 codes for the remainder of the protein, such as the putative 13-zinc-finger DNA binding domain and the predicted ~4 kb 3′UTR. It is possible to splice out the third KRAB-containing exon to produce a shorter in-frame protein encoded by exon 4, i.e., an alternative KRAB-less isoform. Nearly half of the 800 C2H2-type *ZNF* genes in humans contain a single, evolutionarily conserved KRAB region, which could be spiced in alternative fashion [11,12,13]. Our study demonstrated that *ZNF71 KRAB* expression was associated with EMT in cell lines and tumors from NSCLC patients [19]. 

ZNF71 protein expression was not available in the CCLE database [37]. Here, ZNF71 protein expression (GeneTex, Cat. No. GTX116553) was detected in A549 and H441 cell lines in whole cell lysate and nuclear fractions isolated from the same cell lines (Figure 4), consistent with its function as a transcription factor. ZNF71 protein expression was positively correlated with epithelial markers, including E-cadherin and Cytokeratin, and was negatively correlated with mesenchymal markers, including ZEB1 and Vimentin in Western blots. These results are consistent with the observed positive association between elevated ZNF71 protein expression and a good prognosis of NSCLC [10]. 

To illustrate the molecular networks of ZNF71 and its isoforms, the expression of *ZNF71* overall and isoforms was analyzed in the NGS dataset GSE81089 of NSCLC tumor samples (*n* = 197) [38]. Proliferation networks of *ZNF71*, *ZNF71 KRAB*, and *ZNF71 KRAB*-less were constructed, respectively, with the Boolean implication network algorithm. First, genes that had a significant co-expression (*p* < 0.05, *z*-tests) with *ZNF71*, *ZNF71* KRAB, or *ZNF71* KRAB-less in GSE81089 were identified in the genome-scale analysis, respectively. Next, the proliferation genes which meet the criteria were selected based on CRISPR-Cas9/RNAi screening data in CCLE NSCLC cell lines (more details were provided in Appendix A). These proliferation genes were pinpointed in the gene co-expression networks to form the proliferation networks. There were six categories of the constructed proliferation networks: (1) when *ZNF71* was up-regulated or not down-regulated (Figure 5A); (2) when *ZNF71* was down-regulated or not up-regulated (Figure 5B); (3) when *ZNF71* KRAB was up-regulated or not down-regulated (Figure 5C); (4) when *ZNF71* KRAB was down-regulated or not up-regulated (Figure 5D); (5) when *ZNF71* KRAB-less was up-regulated or not down-regulated (Figure 5E); and (6) when *ZNF71* KRAB-less was down-regulated or not up-regulated (Figure 5F). The gene expression signatures containing up- and down-regulated NSCLC proliferation genes from these six networks were input into CMap, respectively, to identify significant (*p* < 0.05) perturbagen signatures and functional pathways. The common significant (connectivity score > 0.9) functional pathways were summarized for *ZNF71* (Table 3), *ZNF71 KRAB* (Table 4), and *ZNF71 KRAB*-less (Table 5). More details were provided in Appendix A. 

The functional pathways associated with the identified proliferation networks of *ZNF71*, *KRAB*, and *KRAB*-less isoforms were largely different, although with some overlaps, i.e., CP_FGFR_INHIBITOR between *ZNF71* and *KRAB*, and PID_CIRCADIAN_PATHWAY for *ZNF71* and KEGG_CIRCADIAN_RHYTHM_MAMMAL for *KRAB*-less. When the up-regulated and down-regulated proliferation genes in each network (Figure 5) were uploaded to CMap separately, significant (*p* < 0.05, connectivity score > 0.9) perturbagen signatures and functional pathways were identified (Appendix A), showing that KD_AHSP_PATHWAY, KD_PHOSPHOLIPASES, and KD_PI3K_SIGNALING were commonly associated with *ZNF71*, *KRAB*, and *KRAB*-less proliferation networks. These results generated hypotheses for future mechanistic studies of *ZNF71* and its isoforms.

### 2.3. Functional Pathways Associated with the Seven-Gene Multi-Omics Network and Discovery of Repositioning Drugs

*ZNF71 KRAB* was reported to be associated with EMT in our previous study [19]. Protein expression of ZNF71 was positively correlated with epithelial markers and negatively correlated with mesenchymal markers in the nuclear fraction of NSCLC cells (Figure 4). The identified seven-gene multi-omics network (Figure 1C) and 14 EMT markers were used to discover connected perturbagen signatures and repositioning drugs with potential implications in improving NSCLC treatment. The following mechanisms of action were considered to reverse EMT, to enhance drug sensitivity, to induce immune response, and to inhibit NSCLC proliferation through (1) up-regulation of epithelial markers; (2) up-regulation of drug sensitive genes; (3) down-regulation of mesenchymal markers; (4) down-regulation of proliferation genes; and (5) down-regulation of *CD27* and *PD-L1* (*CD274*). The 14-gene EMT classifier was used in our previous study [19] to define the EMT states of NSCLC tumors, including three mesenchymal markers (*ZEB1, VIM,* and *FN1*) and 11 epithelial markers (*CDH1*, *EPCAM*, *ESRP1*, *ESRP2*, *DDR1*, *CTNNB1*, *CD24*, *CLDN7*, *KRT8*, *KRT19*, and *RAB25*). The drug sensitive genes were selected if a gene’s mRNA expression was sensitive to one or more of the studied 10 NSCLC therapeutic regimens and was not resistant to any studied drugs in PRISM or GDSC1/2 datasets. The drug sensitive genes and the epithelial markers were excluded from the list if they were also proliferation genes. Here, the proliferation genes referred to those with a significant dependency score (<−0.5) in CRISPR-Cas9 or RNAi in more than 10 NSCLC cell lines. The proliferation gene list also excluded the epithelial markers (Figure 6A). 

With the above defined up- and down-regulated gene lists, significant (*p* < 0.05, connectivity score > 0.9) functional pathways were identified with CMap (Table 6). The details of results were provided in Appendix A, sheet 4. Specifically, shRNA knock-down of the following pathways matched the seven-gene multi-omics network, including KD_APOLIPOPROTEINS, KD_CYCLINS, KD_LYSINE_ACETYLTRANSFERASES, KD_NON_HOMOLOGOUS_END_JOINING, KD_V_TYPE_ATPASES, and OE_NFKB_ACTIVATION (Table 6). After checking the gene co-expression networks in NSCLC patient tumors in GSE31800 and GSE28582, *XRCC5* from KD_NON_HOMOLOGOUS_END_JOINING pathway was found to have the same gene co-expression patterns in patient tumors as in shRNA knock-down in A549 cell line (Figure 6B; Table 6). A co-downregulation between *XRCC5* and *CDCA3*, *ERH,* and *KRT17,* respectively, was found in both NSCLC tumors and LINCS L1000 CMap NSCLC cell lines (Figure 6B). 

Based on the designed mechanisms of action, 39 repositioning drugs were identified with CMap (*p* < 0.05, Appendix A). From this list, drugs that have lower average IC_50_ and EC_50_ in CCLE NSCLC cell lines were selected for potential inhibition of the growth of lung cancer cells with a safer dose. In the original PRISM drug screen, eight doses ranging from 0.0006 µM to 10 µM were tested on most of the compounds [39]. Five small molecules (dasatinib, FK-888, homosalate, lestaurtinib, and penfluridol) had a small average IC_50_ or EC_50_ value (smaller than or close to 1 µM) in the PRISM drug screen and were selected for drug repurposing (Figure 6C, more details provided in Appendix A). Dasatinib is a targeted therapy used to treat chronic myelogenous leukemia and acute lymphoblastic leukemia. FK 888 is a selective, high-affinity tachykinin NK1 receptor antagonist [40]. Homosalate is an organic compound commonly used in sunscreen that absorbs shortwave UVB rays, which helps prevent DNA damage and decreasing the risk of skin cancer. Lestaurtinib is a tyrosine kinase inhibitor and is structurally related to staurosporine. It has undergone clinical trials for the treatment of various cancers, including leukemia [41] and refractory pediatric neuroblastoma [42]. Penfluridol is a first-generation antipsychotic drug used to treat schizophrenia [43]. These results provided evidence for future drug repurposing study to improve NSCLC treatment in combination with clinical therapies.

## 3. Discussion

In both males and females, lung cancer remains the leading cause of cancer-related death. Lung cancer is difficult to manage clinically due to its complex etiology and a lack of biomarkers for the selection of therapy in individual patients. A National Cancer Institute report shows that 56% of lung cancer cases are diagnosed at the distant (metastatic) stage [44]. Novel biomarkers can aid the selection of appropriate chemotherapy and immunotherapy for treatment of advanced NSCLC. For early-stage NSCLC, it remains a challenge to improve patient survival due to the varying performance of lymphadenectomy and insufficient understanding of molecular features harboring recurrence [45]. The development of clinically applicable prognostic biomarkers for early-stage NSCLC could offer a solution by identifying tumors prone to recurrence. Thus, a significant unmet need exists for a reliable diagnostic method to (1) identify stage I NSCLC who might benefit from adjuvant chemotherapy; and (2) select specific therapeutic regimens that a NSCLC patient might respond to.

We sought to meet these critical needs with the development of a qRT-PCR based seven-gene microfluidic assay for early-stage NSCLC prognosis and prediction of benefits of chemotherapy [10]. The seven-gene assay was constructed from the feasibility-training cohort with 83 patients based on accurate patient stratification (*p* = 0.0043), and it has been validated in independent patient cohorts including a clinical trial JBR.10 [46] (*n* = 248, *p* < 0.0001) according to Kaplan–Meier analysis. In both training (*p* = 0.035) and validation (*p* = 0.0049) sets, patients who received adjuvant chemotherapy had significantly better disease-specific survival than patients who did not receive chemotherapy in the predicted chemotherapy-benefit group. Patients receiving chemotherapy in either set did not show a survival benefit in the predicted non-benefit group [10]. The ability of this gene assay to predict the recurrence risk and recommend early implementation of chemotherapy in stage I NSCLC patients can potentially improve patient care. In this study, we showed that it is feasible to apply this seven-gene panel for NSCLC prognosis using NGS profiles of the TCGA patient tumors randomly partitioned into training and testing sets. In future studies, the development of an NGS-based seven-gene assay will need to be defined with specific NGS data generation protocols, data processing methods, and model parameters from the training set that are validated in external patient cohorts for potential clinical applications. 

Within this seven-gene assay, we recently found that *ZNF71 KRAB* isoform is associated with EMT in NSCLC tumors and cell lines, i.e., expressed higher in top 50% of *ZEB1* expressing cells [19]. ZNF71 protein expression was positively correlated with epithelial markers, including E-cadherin and Cytokeratin, and was negatively correlated with mesenchymal markers, including ZEB1 and Vimentin in NSCLC cell lines in Western blots. These results were consistent with the observed positive association between ZNF71 protein expression and prolonged NSCLC patient survival [10]. Together, these results supported the transcriptional repression role of the KRAB domain. 

Approximately 3% of the human genome is composed of zinc finger proteins (ZFPs), which are the largest transcription factor family in human cells and most eukaryotes [47,48]. ZFPs contain a repeating structure of two histidine and two cysteine amino acid residues (i.e., C2H2) that coordinate a zinc ion, which is capable of binding to DNA, RNA, or cellular proteins [49]. A number of biological functions have been predicted for ZFPs as a result of their molecular structure, including DNA repair, protein degradation, signal transductions, cell migration, apoptosis regulation, lipid binding, and transcription regulation [48,49]. The KRAB-ZFPs family of transcriptional repressors plays a variety of roles, most importantly the silencing of transposable elements [50]. KRAB-ZFPs are composed of an N-terminal KRAB domain and a C-terminal C2H2-type zinc finger array. The recruitment of KRAB-associated protein 1, also known as tripartite motif protein 28, KAP1/TRIM28, plays an essential role in the repressor function of KRAB-ZEPs. KAP1/TRIM28 acts as a scaffold complex comprised of histone methyl transferase (SETDB1), heterochromatin protein-1 (HP-1), nucleosome remodeling and deacetylation (NuRD), and DNA methyl transferase [13]. The constructed repressor complex triggers heterochromatin production when KRAB-ZFPs bind KAP1/TRIM28. KAP1/TRIM28 is a universal co-factor for KRAB-ZFP transcription factors [24] and is involved in a variety of cellular functions, ranging from promoting cell proliferation [51] to mediating anti-proliferative activities [52]. The KAP1/TRIM28 protein is involved in tumorigenesis by formatting heterochromatin, mediating DNA damage response, inhibiting *p53* activity, regulating EMT, and maintaining stem cell pluripotency and genome stability [53]. TGF-β treatment induces KAP1/TRIM28 mRNA and protein expression. KAP1/TRIM28 deficiency blocks TGF-β induced EMT and reduces cell migration and invasion. KAP1/TRIM28 regulates E-cadherin and N-cadherin promoters, implying that KAP1/TRIM28 contributes to EMT and is involved in lung cancer metastasis [25]. Upregulation of RB-associated KRAB zinc finger (BRAK) and its association with poor prognosis in NSCLC was reported [51]. Zinc finger protein 668 (ZNF668) down-regulates Snail and upregulates E-cadherin and zonula occludens-1, which, in turn, suppresses NSCLC invasion and migration [54]. ZNF668 protein expression was downregulated in tumors compared with normal lung tissues and was negatively associated with lymph node metastasis in NSCLC [54]. 

The role of *ZNF71* and its isoforms in EMT is unclear. EMT may be linked to significant splicing changes, and as a consequence, it could potentially skew splicing for inclusion of the KRAB domain [55,56]. To better understand the functions of *ZNF71* and its isoforms, genome-scale co-expression networks containing all the genes with a statistically significant co-expression association relation with *ZNF71* overall, *KRAB*, and *KRAB*-less were constructed, respectively. Proliferation networks were dissected based on CRISPR-Cas9/RNAi screening data in CCLE NSCLC panel. Significant functional pathways matching the up- and down-regulation signatures of these proliferation networks were identified with L1000 CMap. The results showed that multiple different functional pathways were associated with *ZNF71*, *KRAB*, and *KRAB*-less isoforms in shRNA knock-down or overexpression assays in NSCLC cell lines, respectively. Three pathways, including KD_AHSP_PATHWAY, KD_PHOSPHOLIPASES, and KD_PI3K_SIGNALING in shRNA knock-down assays of NSCLC cell liens, were commonly associated with *ZNF71*, *KRAB*, and *KRAB*-less isoforms. These results provided hypotheses of future molecular and cellular studies of *ZNF71* and its isoforms in NSCLC.

It has been found that the seven-gene panel interacted with major inflammatory and cancer signaling hallmarks including *TNF*, *PI3K*, *NF-κB*, and *TGF-β* [10]. The protein expression of CD27 quantified with ELISA had a strong correlation with its mRNA expression in NSCLC tumors [10]. As a new generation of immunotherapy target [57,58], CD27 is currently being tested in phase I/II clinical trials for multiple tumor types with promising results [59,60]. The treatment effectiveness of cancer vaccines and immunotherapy can be improved by CD27 agonist antibodies, either alone or in combination with anti-PD1 antibodies [61,62]. There was reported synergy between PD-1 blockade and CD27 stimulation for CD8+ T-cell driven anti-tumor immunity [62], indicating the therapeutic potential of CD27 in neoadjuvant PD-1 blockade in early-stage NSCLC [63]. 

Novel biomarker discovery is complicated by combinations of chemotherapy and immunotherapy, and multiple intricate interactions relevant to immune responses should be considered in this process [64]. The NSCLC tumor immune microenvironment can be represented as a multi-dimensional component network of immune and stromal cells as well as blood vessels, interacting with epithelial tumor cells in a convoluted manner. An integrated analysis of genome-scale CNV and gene expression profiles in NSCLC tumors was conducted using the Boolean implication network methodology to model this complicated molecular machinery [65]. The presented Boolean implication network algorithm is conceptually innovative and can model molecular circuits, such as feedback loops, which are important in biological switches and signaling motifs that regulate large-scale cellular processes [66]. The multi-omics network involving the seven-gene signature was dissected. All the network genes, including *PD-L1 (CD274)*, had a significant correlation with immune infiltration in at least one immune cell type of TCGA NSCLC tumors, including myeloid dendritic cells, macrophages, neutrophils, CD4+ T cells, CD8+ T cells, and B cells. Among the network genes, down-regulation of *SERPINB5* by small interfering RNA increased the resistance of NSCLC H460 cells to ionizing radiation as reported in the literature [67], consistent with the observed association between *SERPINB5* up-regulation and prolonged survival in TCGA NSCLC patients receiving radiotherapy. Low XRCC5 protein expression was more prevalent among squamous cell carcinoma compared with lung adenocarcinoma compared with normal lung tissues in Lee et al. [68]. The cell division cycle-associated gene (CDCA3), mRNA, and protein expressions were increased in NSCLC compared with normal tissue, and high levels of CDCA3 was associated with poor prognosis in Adams et al. [69].The mRNA and protein expressions of numerous network genes were associated with drug sensitivity or resistance to 10 commonly used therapeutic regimens for treating NSCLC. These results further substantiate the potential clinical implications of the multi-omics network of the seven-gene signature. 

The multi-omics network genes had a higher percentage (82.9%) of correlated mRNA and protein expression in CCLE NSCLC cell lines compared with the rest of the genome (66.3%). The correlation between mRNA and protein expression levels in complex biological systems is in general poor [70]. Many studies reported dysregulated gene expression as evidence of the consequent dysregulation of protein expression. Nevertheless, this is not necessarily true in many circumstances. Current RNA sequencing and reverse-phase protein array technologies have revealed genome-scale mRNA and protein expression and their correlation to better understand the transcriptional and translational processes. When reverse-phase protein arrays or commercial antibodies are not available for a research project, the mRNA expression can be used as a surrogate of the protein expression for genes with highly correlated mRNA and protein expression levels.

Molecular pathway and network approaches are promising for the discovery of novel therapeutic targets and repositioning drug candidates [71]. Drug repositioning is essential for pharmaceutical research and development (R & D) due to the proven toxicity profiles [71]. To design therapeutic strategies to reverse EMT, enhance drug sensitivity, and inhibit NSCLC proliferation, significant functional pathways and repositioning drugs were identified with L1000 CMap. The 14-gene EMT classifier to characterize NSCLC tumors [10], the multi-omics network of the seven-gene signature, and CRISPR-Cas9/RNAi screening data were used to generate up- and down-regulated gene lists as input to CMap. ShRNA knock-down of the following pathways were linked to these mechanisms, including KD_APOLIPOPROTEINS, KD_CYCLINS, KD_LYSINE_ACETYLTRANSFERASES, KD_NON_HOMOLOGOUS_END_JOINING, KD_V_TYPE_ATPASES, and OE_NFKB_ACTIVATION. Nuclear factor kappa B (NF-ƙB) is a master regulator in NSCLC pathogenesis and progression [72,73]. Activated NF-ƙB signaling may influence the progression of a lung tumor either positively via triggering immune surveillance [74] or negatively via promoting inflammation and tumorigenesis [75]. Activation of NF-ƙB stimulates transcription of cyclin D1, a key regulator of G1 checkpoint control [76]. NNK, a tobacco-specific nitrosamine, can stimulate proliferation through activation of the NF-ƙB and subsequent upregulation of cyclin D1 in a normal human bronchial cell [77]. Inhibition of NF-ƙB results in loss of induction of EMT master-switch transcription factors in NSCLC [78]. NF-ƙB activation has been associated with resistance to chemotherapeutics and tyrosine kinase inhibitors in EGFR mutant lung cancer [72]. Protein expression of V-ATPase is associated with drug resistance and higher tumor grade in NSCLC [79]. Dasatinib and lestaurtinib used for leukemia treatment were identified as repurposing drugs for NSCLC. Homosalate, FK 888, and penfluridol were discovered as potential repositioning drugs that can kill NSCLC cells, maintaining the expression of drug sensitivity genes and epithelial markers, and inhibit mesenchymal markers, proliferation genes, and ICIs for NSCLC treatment. We developed a novel pipeline to identify repositioning drugs based on 1) network-based identification of genes with prognostic and chemo-predictive implications, 2) pre-selection of repositioning drug candidates using CMap, and 3) further selection of repositioning drugs that can effectively kill NSCLC cells with the PRISM and GDSC1/2 drug activity databases. The pipeline presented in this study can aid pharmaceutical R & D in determining disease relevance and designing of clinical trials of repositioning drugs to minimize the risk of failures, to expedite drug development, and to reduce costs.

## 4. Materials and Methods

### 4.1. Patient Cohorts

#### 4.1.1. TCGA NSCLC Patient Cohorts

The TCGA-LUAD (*n* = 515) and TCGA-LUSC (*n* = 501) data were obtained from LinkedOmics [80] (http://linkedomics.org/, accessed on 28 April 2021). Normalized gene-level reverse phase protein array data, gene expression, and patient clinical information were available from the source.

#### 4.1.2. NSCLC Patient Cohort GSE31800

A previous study [81] measured DNA copy number profiles in 271 samples from NSCLC tumors (GSE31800). A total of 49 samples had matched custom microarray gene expression profiles [81]. 

#### 4.1.3. NSCLC Patient Cohort GSE28582

This study included a publicly available NSCLC patient cohort (*n* = 100) [82,83] (GSE28582). All 100 samples had both SNP array profiled DNA copy number variation and microarray gene expression data. 

#### 4.1.4. NSCLC Patient Cohort GSE81089

A total of 197 patients with sufficient survival information out of 199 NSCLC patient tumor samples from GSE81089 [38] were included in this study. The transcripts per million (TPM) reads of *ZNF71* at the isoform level were generated from the raw RNA-seq data with the Salmon method [84].

### 4.2. Data Pre-Processing

#### 4.2.1. CNV Data Pre-Processing

The CNV data from GSE31800 [81] and GSE28582 [82,83] cohorts were matched with the converted Hg38 chromosome locations. Details of the CNV data pre-processing were provided in our previous study [65]. The pre-processed CNV data were categorized into 3 levels: 1 (amplification), −1 (deletion), and 0 (normal).

#### 4.2.2. Gene Expression Data Pre-Processing

In order to categorize the level of gene expression, we used 27 housekeeping genes (*ACTB*, *B2M*, *CDKN1B*, *ESD*, *FLOT2*, *GAPDH*, *GRB2*, *GUSB*, *HMBS*, *HPRT1*, *HSP90AB1*, *IPO8*, *LDHA*, *NONO*, *PGK1*, *POLR2A*, *PPIA*, *PPIH*, *PPP1CA*, *RHOA*, *RPL13A*, *SDCBP*, *TBP*, *TFRC*, *UBC*, *YAP1*, and *YWHAZ*) [10,85,86,87,88] to define the thresholds of gene expression level. The total percentage of over-expression and under-expression samples for all the housekeeping genes was fixed to be 30%, and the number *n* that meets the following Equation (1) was computed for each gene [88,89]: (1)$$\begin{array}{l} \left(number\;of\;samples\;have\;expression>mean+n\times std\right)\\ \;+\left(number\;of\;samples\;have\;expression<mean-n\times std\right)\\ \;=30\%\;of\;total\;number\;of\;samples \end{array}$$

The averaged *n* value of all 27 housekeeping genes was 0.874 in the gene expression data of GSE31800 and 0.977 in the gene expression data of GSE28582, respectively. These *n* values were used to categorize each gene’s expression in each dataset. Details of gene expression normalization using these housekeeping genes were provided in our previous publication [65]. 

### 4.3. Boolean Implication Networks

In the present study, an implication network algorithm based on prediction logic [90] was used. This algorithm was proposed by Guo et al. [91,92]. Details of the implication network algorithm and its application in CNV and gene expression analysis were provided in our previous study [65]. For an implication rule to be considered significant, it must have a scope and precision values that are greater than the threshold values, as well as being the largest among all the rules for the specific pair of variables. Based on the significance level of statistical tests, the threshold for scope and precision will vary. Based on the sample size, the threshold value was calculated using a one-tailed *z*-test using the ideal value of *z*. The *z* value used for this study was 1.64 (95% confidence interval, α = 0.05, one-tailed *z*-tests) [65].

### 4.4. Cancer Cell Line Encyclopedia (CCLE)

CCLE cell line RNA-seq data were obtained from DepMap 20Q2 (https://figshare.com/articles/dataset/DepMap_20Q2_Public/12280541, accessed on 1 April 2021) [93]. Gene expression data quantified with the GTEx pipelines [94] were obtained from the CCLE data portal (https://data.broadinstitute.org/ccle/CCLE_RNAseq_081117.rpkm.gct, accessed on 1 April 2021). A total of 117 NSCLC cell lines were included in this analysis.

Proteomics data for CCLE were downloaded from the Gygi lab (https://gygi.hms.harvard.edu/publications/ccle.html, accessed on 1 October 2021) [37]. A panel of 63 NSCLC cell lines were included in this analysis.

### 4.5. CRISPR-Cas9 Knockout Assays

Project Achilles has quantified CRISPR-Cas9 gene knockout effects in CCLE [95,96] available in version DepMap 20Q2 (https://figshare.com/articles/dataset/DepMap_20Q2_Public/12280541, accessed on 1 April 2021) [93]. CERES [95] was used to process the screening data for CRISPR-Cas9. Genes are defined as essential if their knockout significantly impact the cellular growth in a cell line; otherwise, they are defined as non-essential. In each cell line, gene knockout effects were normalized so that the median non-essential gene knockout effect was 0, and the median essential gene knockout effect was –1. Genome-scale CRISPR-Cas9 knockout results in 78 NSCLC cell lines were included in this study. If a gene has a normalized dependency score less than –0.5 in a sample, it is considered to have a significant effect in CRISPR-Cas9 knockout.

### 4.6. RNAi Knockdown Assays

Project Achilles also contains genome-scale RNAi screening data of CCLE (https://depmap.org/R2-D2/, accessed on 1 April 2021) [97]. The average gene dependency scores in each cell line for short hairpin RNA (shRNA) libraries were estimated using the DEMETER2 method [97]. The gene dependency scores were standardized such that the median of the across-cell-line average dependency scores of the positive control gene set was –1, and the median of the averaged scores of the negative control gene set was 0. Same as CRISPR-Cas9, if a sample has normalized dependence score less than –0.5, that indicates it has a significant effect in RNAi knockdown. A total of 92 NSCLC cell lines were included in the RNAi analysis.

### 4.7. Immune Infiltration Estimation

TIMER 2.0 [34,35,36] was used to compute the association of gene expression and immune infiltration in multiple cell types using a variety of immune deconvolution methods.

### 4.8. PRISM Drug Response in CCLE

The PRISM molecular barcoding and multiplexed screening method quantified the growth inhibitory activity of 4518 drugs in 578 human cancer cell lines [39]. The PRISM dataset was obtained from the Cancer Dependency Map portal (https://depmap.org/portal/download/, accessed on 1 April 2021). In this study, we focused on the activities of nine drugs for treating NSCLC, including carboplatin, cisplatin, paclitaxel, docetaxel, gemcitabine, vinorelbine, etoposide, gefitinib, and erlotinib. In the experiments, different levels of doses were designed for the treatments by each drug. A cell line was directly defined as resistant if its IC_50_ or EC_50_ value is higher than the maximum dose; a cell line was defined as sensitive if its IC_50_ or EC_50_ value is lower than the minimum dose. The remaining cell lines with IC_50_ or EC_50_ value in-between the minimum and maximum doses were divided into groups of resistant, sensitive, or partial response. Details of drug response categorization were provided in our previous studies [19,65].

### 4.9. Genomics of Drug Sensitivity in Cancer (GDSC1/2)

Drug screening data were available from the Genomics of Drug Sensitivity in Cancer (GDSC) Project (https://www.cancerrxgene.org/downloads/bulk_download, accessed on 15 April 2021) [98]. In more than 1000 human cancer cell lines, a broad spectrum of anti-cancer therapeutic compounds was screened in the GDSC Project. Details of GDSC1/2 and drug response categorization were described in our previous publications [19,65].

### 4.10. Drug Repurposing Using Connectivity Map (CMap)

CMap uses transcriptional expression signatures to connect disease, cell physiology, and therapeutics. CMap contains a library of over 1.5M mRNA expression profiles from ~5000 small-molecule compounds and ~3000 genetic reagents, screened in multiple cell types. Using these screening data, CMap generates hypotheses of functional pathways and repositioning drugs through the matching of gene expression signatures containing up-regulated and/or down-regulated gene lists as database queries. CMap (https://clue.io/, accessed on 1 October 2021) [26,27] was used to find perturbagen signatures and functional pathways that connected with the input gene expression signatures.

### 4.11. Western Blots

NSCLC cell lines H441, H23, and H1299 (kind gift of Dr. Scott Weed, WVU), A549 and H1975 (kind gift of Dr. Ivan Martinez, WVU) were grown in DMEM (Corning, cat. # 15-018-CV) supplemented with 10% FBS (HyClone, UT, USA), 2mM L-glutamine (Corning, Steuben County, NY, USA; cat. # 25-005-CI), and 1 × Antibiotic Antimycotic Solution (Corning, Steuben County, NY, USA; cat. # 30-004-CI). All cells were maintained at 37 °C in a 5% CO_2_ incubator. Whole/total cell lysates were prepared in nonreducing Laemmli buffer as described in [99]. Nuclear-cytoplasmic fractionation was performed using ProteoJET Cytoplasmic and Nuclear Protein Extraction Kit (Fermentas, cat. # K0311) as described in [51].

Protein concentration was quantified by Pierce BCA Protein Assay (ThermoFisher, Waltham, MA, USA; cat # 23225). Lysates with equal amount of total protein were separated on 4% to 12% Bis-Tris NuPAGE Novex gels (Invitrogen, Waltham, MA, USA) and transferred to a polyvinylidene difluoride (PVDF) membrane. Protein bands were detected using standard chemiluminescence techniques using GE Healthcare Amersham Imager 680.

ZNF71 antibody was purchased from GeneTex (Cat. No. GTX116553). The following antibodies were included in Western blots: ZEB1 (Sigma, St. Louis, MO, USA; HPA027524), Vimentin (V-9; Santa Cruz Biotechnology, Dallas, TX, USA; sc-6260), E-cadherin (BD Biosciences, Franklin Lakes, NJ, USA; cat# 610181), pan-Cytokeratin (C11, Santa Cruz Biotechnology, Dallas, TX, USA; sc-8018), PCNA (eBioSciences, 14-6748-81), and Tubulin α (Sigma, St. Louis, MO, USA; T9026). Standard chemiluminescence was used to detect the protein bands.

### 4.12. Statistical Methods

Rstudio version 1.4.1106 [100] was used as the main software in the statistical analysis of this study. Student’s *t*-tests were used to assess differences in mRNA and protein expression between two groups; statistical significance was defined as having a two-sided *p*-value < 0.05. The multivariate Cox regression analysis was used to compute risk-scores. Kaplan–Meier analysis was applied with the survival package in R to perform the survival analysis. In Kaplan–Meier analyses, log-rank tests were used to determine the difference in survival probability between different groups. The relationship between two sample groups were evaluated with Pearson’s correlation test. One-tailed two proportion *z*-test was used to compare if two proportions are the same.

## 5. Conclusions

This study substantiated the feasibility of applying the seven-gene signature developed with qRT-PCR microfluidic assay to TCGA NGS data for NSCLC prognosis. CNV mediated transcriptional regulatory network of the seven-gene signature was identified in multiple NSCLC cohorts using Boolean implication networks. The multi-omics network genes, including *PD-L1 (CD274)*, were significantly correlated with immune infiltration and drug sensitivity or resistance to 10 commonly used therapeutic regimens for treating NSCLC. Dasatinib, lestaurtinib, homosalate, FK 888, and penfluridol were discovered as potential repositioning drugs that can kill NSCLC cells, maintain the expression of drug sensitivity genes and epithelial markers, and inhibit mesenchymal markers, proliferation genes, and ICIs for NSCLC treatment. ShRNA knock-down of NF-ƙB and cyclin D1 pathways were identified as associated with the multi-omics network for designing therapeutic strategies for NSCLC intervention. PI3K was identified as a potential pathway related to proliferation networks involving *ZNF71* and its isoforms for future mechanistic study. We acknowledge that the identified functional pathways and repositioning are based on the matching of the gene expression signatures with screening data in the CMap databases. Although shRNA knockdown and overexpression experimental results were included in the similarity matching in the CMap queries, these results from bioinformatics analysis will need to be further validated in molecular and cellular experiments. 

## 6. Patents

The seven-gene microfluidic assay was filed under US Non-Provisional Patent Pub. No. US 2021-0254173 A1. The results in this study were filed under US provisional patent application 63/265,649.

## Figures and Tables

**Figure 1 ijms-23-00219-f001:**
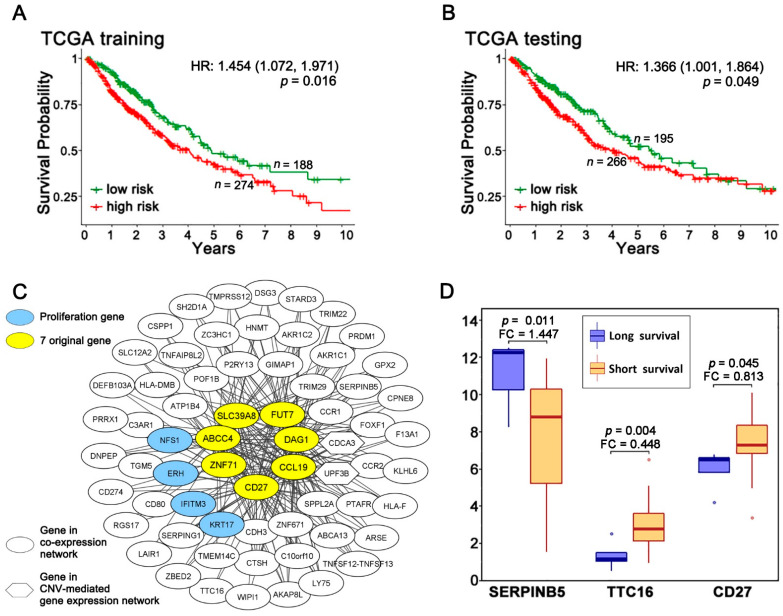
The 7-gene signature and associated multi-omics network. Kaplan–Meier analysis of 7-gene signature in training (**A**) and testing (**B**) sets randomly partitioned from the combined TCGA-LUSC and TCGA-LUAD data. Patients were grouped by risk-scores with the cutoff value of −0.74. The plots show the survival outcome of the first 10 years after surgery. (**C**) The multi-omics network of the 7-gene signature in NSCLC tumors. All proliferation genes in the networks were identified from CRISPR-Cas9/RNAi screening data in CCLE NSCLC cell lines, with more details provided in Appendix A. (**D**) Genes associated with radiotherapy response. In TCGA-LUSC and TCGA-LUAD, these genes showed significantly different expression levels (*p* < 0.05, two-sample *t*-tests) between the long survival group (>58 months; *n* = 144) and the short survival group (<20 months; *n =* 186). Analysis was conducted on stage III or IV patients who had received radiotherapy.

**Figure 2 ijms-23-00219-f002:**
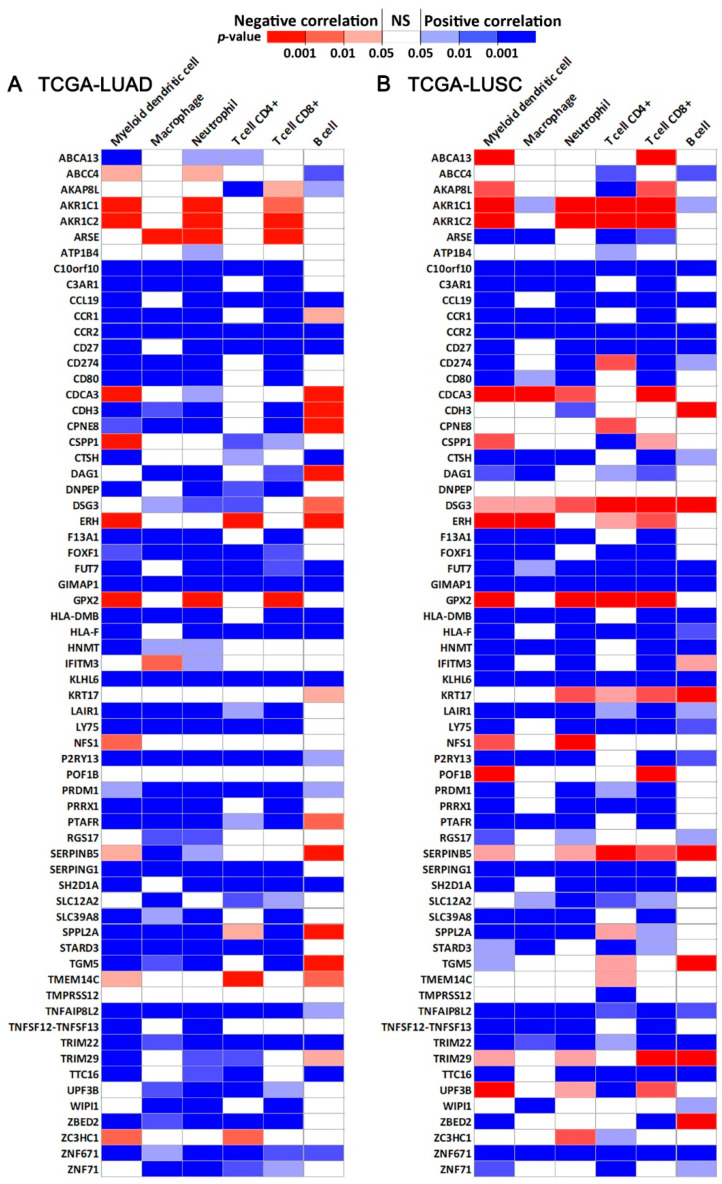
Association between immune infiltration and expression of the multi-omics network genes (Figure 2C). (**A**) The correlation of gene expression with immune infiltration level in TCGA-LUAD patients (*n* = 515) assessed with TIMER 2.0. (**B**) The correlation of gene expression with immune infiltration level in TCGA-LUSC patients (*n* = 501) assessed with TIMER 2.0. NS: not statistically significant.

**Figure 3 ijms-23-00219-f003:**
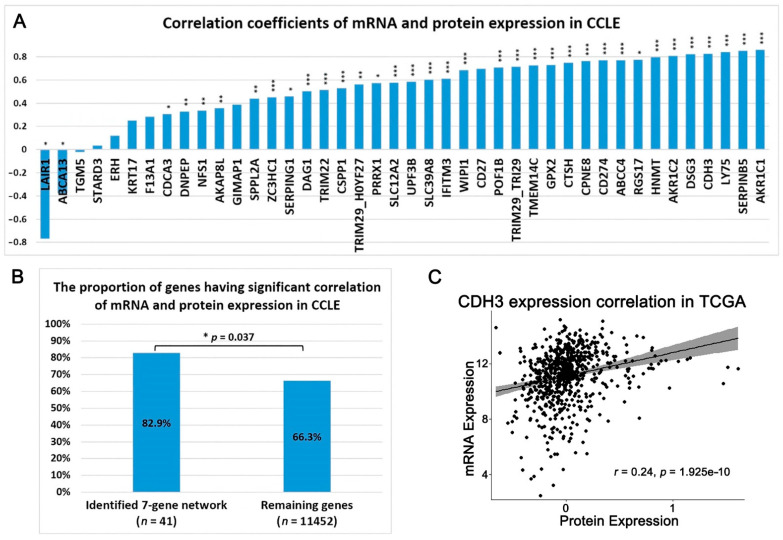
The correlation between mRNA and protein expression of genes in the multi-omics network (shown in Figure 1C). (**A**) The bar chart shows the correlation coefficients of mRNA and protein expression from the same gene in NSCLC cell lines. The asterisk (*) indicates the significance level (* *p* < 0.05, ** *p* < 0.01, *** *p* < 0.001). (**B**) The proportion of genes having a significant correlation of mRNA and protein expression in NSCLC cell lines (*p* < 0.05, Pearson’s correlation). The proportion of genes having a significant correlation between mRNA and protein expression is significantly higher in identified 7-gene network than the remaining genes in the genome (two-tailed proportion *z*-test, *p* < 0.05). (**C**) CDH3 had a significant positive correlation between mRNA and protein expression in combined TCGA-LUSC and TCGA-LUAD patient tumors (*n* = 685).

**Figure 4 ijms-23-00219-f004:**
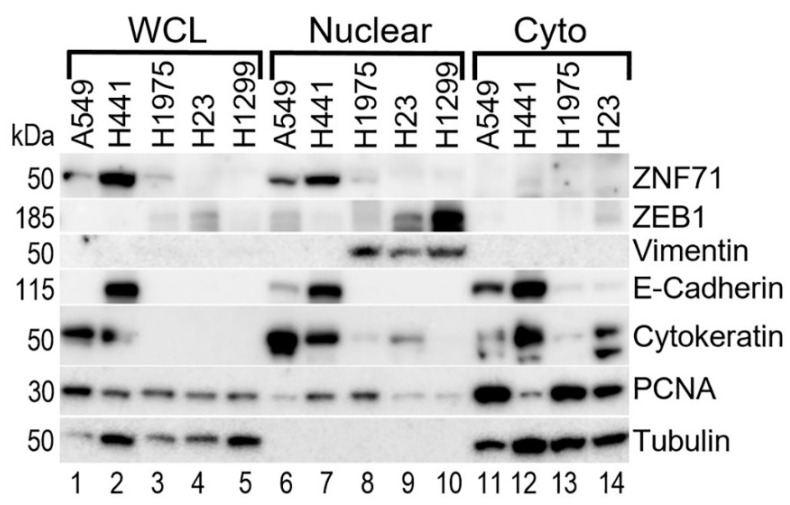
Correlation of expression of ZNF71 with EMT markers in five NSCLC cell lines. Western blotting analysis of *ZNF71*, mesenchymal and epithelial markers in the indicated cell lines. PCNA and Tubulin—loading controls. WCL—whole cell lysate, Nuclear and Cyto—nuclear and cytoplasmic fractions isolated from the same cell lines.

**Figure 5 ijms-23-00219-f005:**
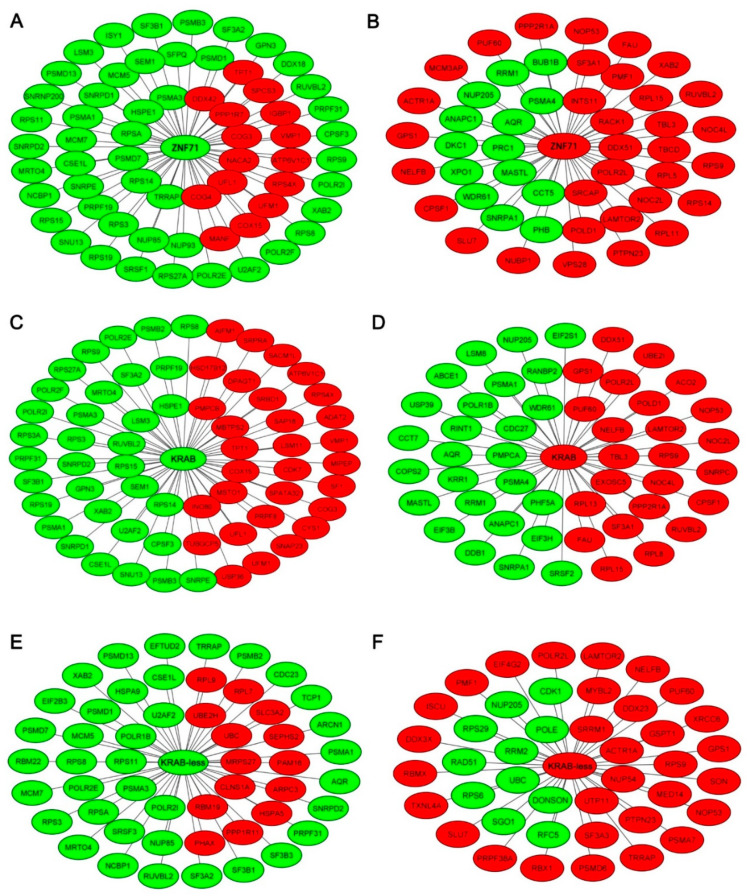
*ZNF71* gene level, *ZNF71* KRAB, and *ZNF71* KRAB-less co-expression networks in patient cohort GSE81089. The green color indicates up-regulation or non-downregulation, and the red color indicates downregulation or non-upregulation. All proliferation genes in the networks were identified from CRISPR-Cas9/RNAi screening data in CCLE NSCLC cell lines, with more details provided in Appendix A. (**A**) Proliferation genes that had a significant (*p* < 0.05, *z*-tests) co-expression with up-regulated *ZNF71*. (**B**) Proliferation genes that had a significant (*p* < 0.05, *z*-tests) co-expression with down-regulated *ZNF71*. (**C**) Proliferation genes that had a significant (*p* < 0.05, z-tests) co-expression with up-regulated ZNF71 KRAB isoform. (**D**) Proliferation genes that had a significant (*p* < 0.05, *z*-tests) co-expression with down-regulated *ZNF71* KRAB isoform. (**E**) Proliferation genes that had a significant (*p* < 0.05, *z*-tests) co-expression with up-regulated *ZNF71* KRAB-less isoform. (**F**) Proliferation genes that had a significant (*p* < 0.05, *z*-tests) co-expression with down-regulated *ZNF71* KRAB-less isoform. Details provided in Appendix A.

**Figure 6 ijms-23-00219-f006:**
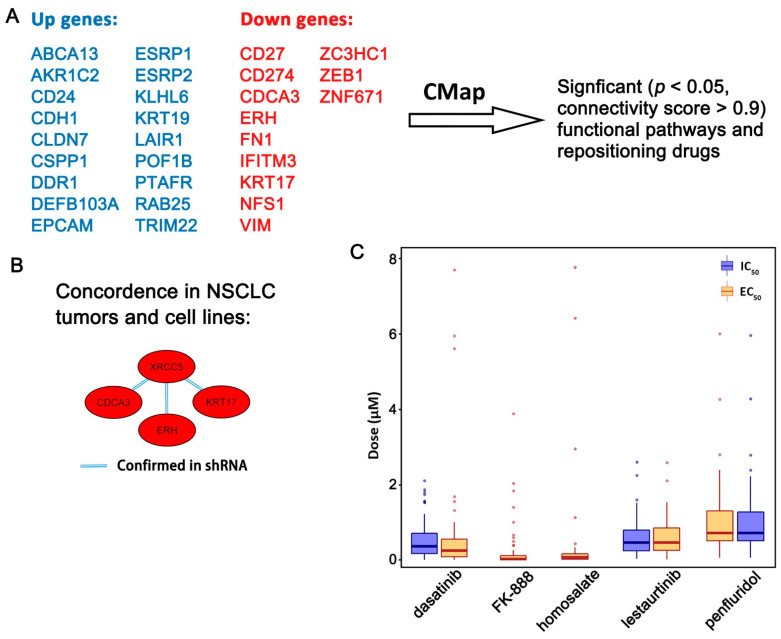
Identification of functional pathways associated with the 7-gene multi-omics network and discovery of repositioning drugs. (**A**) Selection of significant functional pathways and repositioning drugs based on the identified 7-gene multi-omics proliferation network with CMap. (**B**) Confirmed co-expression network of *XRCC5* in NSCLC patient tumors in GSE31800 or GSE28582 showing concordant co-expression patterns after shRNA knock-down of *XRCC5* in A549 cell line. The red color indicates down-regulation or non-upregulation of genes. The double line indicates that the co-expression relations (*p* < 0.05, *z*-tests) observed in patient tumors were confirmed in shRNA knockdown experiments in LINCS L1000 CMap NSCLC cell lines. (**C**) Small molecules that had a low average concentration of drug response in the CCLE NSCLC cell lines. Measurements with a drug activity value larger than 10 μM were considered outliers and were removed from the plot. Dasatinib EC_50_ measurements had 2.38% of outliers. FK-888 EC_50_ measurements had 14.49% of outliers. Homosalate EC_50_ measurements had 7.25% of outliers. Penfluridol had 1.25% of outliers in both IC_50_ and EC_50_ measurements. Detailed information was provided in Appendix A.

**Table 1 ijms-23-00219-t001:** Genes (shown in Figure 1C) with a significant differential mRNA expression (*p* < 0.05; two sample *t*-tests) and a fold change < 0.5 or >2 in sensitive versus resistant CCLE NSCLC cell lines (*n* = 117) to the studied drugs. Blue font indicates higher mRNA expression in sensitive cell lines, and red font indicates high mRNA expression in resistant cell lines.

Drug	PRISM	GDSC1	GDSC2
**Carboplatin**	** *IFITM3, ATP1B4, PRRX1, PRRX1* **		
**Cisplatin**	** * FUT7 * **		** *POF1B, ZNF671, ZBED2, C10orf10* **
**Docetaxel**	** *ERH, KLHL6* **	** *KLHL6, IFITM3, C10orf10* **	** * SERPING1 * **
**Erlotinib**	** *KLHL6, TRIM29, IFITM3, CCR2, TNFSF12-TNFSF13, ATP1B4* **	** *IFITM3, SERPINB5, PRDM1, RGS17, CD80, AKR1C1, TRIM29* **	** *DEFB103A, CSPP1, TRIM29, SERPINB5, AKR1C2, KLHL6, ATP1B4, SERPING1, DSG3* **
**Etoposide**	** *PRDM1, TRIM29* **	** *ATP1B4, SERPING1, ABCA13, ARSE* **	
**Gefitinib**	** *DEFB103A, PTAFR, ZEB1, ERH, F13A1* **	** *DEFB103A, KLHL6, TRIM29, RGS17, ATP1B4, PRRX1* **	** *CSPP1, KLHL6, TRIM29, ATP1B4, SERPING1, ARSE* **
**Gemcitabine**	** *ARSE, GPX2, RGS17, LAIR1, TRIM22* **	** * SERPING1 * **	** *ARSE, C10orf10* **
**Paclitaxel**	** *PTAFR, TRIM29, RAB25, ESRP1 SERPINB5, KLHL6, CD80, SH2D1A* **	** * CD27 * **	** * C10orf10 * **
**Pemetrexed**		** * SH2D1A * **	
**Vinorelbine**	** *FUT7, TNFSF12-TNFSF13, DSG3* **		

**Table 2 ijms-23-00219-t002:** Genes (shown in Figure 1C) with a significant differential protein expression (*p* < 0.05; two sample *t*-tests) and a fold change < 0.5 or >2 in sensitive versus resistant CCLE NSCLC cell lines (*n* = 63) to the studied drugs. Blue font indicates higher protein expression in sensitive cell lines, and red font indicates higher protein expression in resistant cell lines.

Drug	PRISM	GDSC1	GDSC2
**Carboplatin**	** * TGM5 * **		
**Cisplatin**	** * TRIM29 * **	** *CD274, HLA-F* **	** *LY75, PRRX1, F13A1, CTSH* **
**Docetaxel**	** * SERPINB5 * **	** *AKR1C2, SERPINB5* **	** *AKR1C2, AKR1C1, CPNE8* **
**Erlotinib**	** *ARSE, HLA-DMB, TRIM22, LY75, TGM5* **	** * AKR1C1 * **	** *POF1B, SERPINB5, F13A1* **
**Etoposide**	** *HLA-F, ABCA13, TGM5* **	** * SERPINB5 * **	
**Gefitinib**		** * SLC39A8 * **	
**Gemcitabine**	** *TGM5, HNMT, CTSH, HLA-F, SERPINB5, CDH3, HNMT, ABCA13, AKR1C2, GPX2, AKR1C1* **	** *SERPINB5, AKR1C1* **	** * AKR1C1 * **
**Paclitaxel**	** *SERPINB5, CDH3, HLA-F, SLC39A8* **		** *PRRX1, LY75, CTSH, TRIM29* **
**Pemetrexed**			
**Vinorelbine**		** *SERPINB5, AKR1C2, AKR1C1, F13A1, HLA-F* **	** * CTSH * **

**Table 3 ijms-23-00219-t003:** The common significant (*p* < 0.05, connectivity score > 0.9) functional pathways from CMap using *ZNF71* up-regulated network genes (Figure 5A) and *ZNF71* down-regulated network genes (Figure 5B) as input.

src_set_id	*ZNF71* Upregulated Network Genes		*ZNF71* Downregulated Network Genes	
	Cell Line	Type	Cell Line	Type
BIOCARTA_CTL_PATHWAY	A549	TRT_SH.CGS	A549	TRT_SH.CGS
	HCC515	TRT_SH.CGS	HCC515	TRT_SH.CGS
CP_FGFR_INHIBITOR (PD-173074, dovitinib, brivanib, orantinib)	A549	TRT_CP	A549	TRT_CP
KD_RNA_POLYMERASE_ENZYMES (*POLR2A*, *POLR2C*, *POLR2D*, *POLR2E*, *POLR2F*, *POLR2I*, *POLR2K*)	A549	TRT_SH.CGS	HCC515	TRT_SH.CGS
KEGG_GALACTOSE_METABOLISM	HCC515	TRT_SH.CGS	HCC515	TRT_SH.CGS
PID_CIRCADIAN_PATHWAY	A549	TRT_SH.CGS	A549	TRT_SH.CGS
PID_INTEGRIN2_PATHWAY	HCC515	TRT_SH.CGS	A549	TRT_SH.CGS
PID_S1P_S1P2_PATHWAY	HCC515	TRT_SH.CGS	HCC515	TRT_SH.CGS
REACTOME_CHYLOMICRON_MEDIATED_LIPID_TRANSPORT	HCC515	TRT_SH.CGS	HCC515	TRT_SH.CGS
REACTOME_JNK_C_JUN_KINASES_PHOSPHORYLATION_AND_ACTIVATION_MEDIATED_BY_ACTIVATED_HUMAN_TAK1	HCC515	TRT_SH.CGS	HCC515	TRT_SH.CGS
REACTOME_PURINE_RIBONUCLEOSIDE_MONOPHOSPHATE_BIOSYNTHESIS	A549	TRT_SH.CGS	A549	TRT_SH.CGS
			HCC515	TRT_SH.CGS
SULFONYLUREA (glipizide, glibenclamide, gliquidone)	HCC515	TRT_CP	HCC515	TRT_CP

**Table 4 ijms-23-00219-t004:** The common significant (*p* < 0.05, connectivity score > 0.9) functional pathways from CMap using *ZNF71* KRAB upregulated network genes (Figure 5C) and *ZNF71* KRAB downregulated network genes (Figure 5D) as input.

src_set_id	KRAB Upregulated Network Genes		KRAB Downregulated Network Genes	
	Cell Line	Type	Cell Line	Type
BIOCARTA_CTL_PATHWAY	HCC515	TRT_SH.CGS	HCC515	TRT_SH.CGS
			A549	TRT_SH.CGS
BIOCARTA_GLYCOLYSIS_PATHWAY	HCC515	TRT_SH.CGS	A549	TRT_SH.CGS
BIOCARTA_SET_PATHWAY	A549	TRT_OE	A549	TRT_OE
CP_CCK_RECEPTOR_ANTAGONIST (devazepide, LY-225910, SR-27897)	HCC515	TRT_CP	HCC515	TRT_CP
CP_FGFR_INHIBITOR (PD-173074, dovitinib, brivanib, orantinib)	A549	TRT_CP	HCC515	TRT_CP
KD_INTEGRIN_SUBUNITS_BETA (*ITGB1*, *ITGB4*, *ITGB5*)	A549	TRT_XPR	A549	TRT_XPR
PID_DNA_PK_PATHWAY	A549	TRT_SH.CGS	A549	TRT_SH.CGS
REACTOME_NFKB_IS_ACTIVATED_AND_SIGNALS_SURVIVAL	A549	TRT_SH.CGS	HCC515	TRT_SH.CGS
REACTOME_REGULATION_OF_COMPLEMENT_CASCADE	HCC515	TRT_SH.CGS	HCC515	TRT_SH.CGS
SODIUM/GLUCOSE_COTRANSPORTER_INHIBITOR (maackiain, phloretin)	A549	TRT_CP	A549	TRT_CP
SULFONYLUREA (glipizide, glibenclamide, gliquidone)	HCC515	TRT_CP	HCC515	TRT_CP
TGF_BETA_RECEPTOR_INHIBITOR	HCC515	TRT_CP	HCC515	TRT_CP

**Table 5 ijms-23-00219-t005:** The common significant (*p* < 0.05, connectivity score > 0.9) functional pathways from CMap using *ZNF71* KRAB-less upregulated network genes (Figure 5E) and *ZNF71* KRAB-less downregulated network genes (Figure 5F) as input.

src_set_id	KRAB-Less Upregulated Network Genes		KRAB-Less Downregulated Network Genes	
	Cell Line	Type	Cell Line	Type
ABL_KINASE_INHIBITOR	A549	TRT_CP	HCC515	TRT_CP
BIOCARTA_BLYMPHOCYTE_PATHWAY	A549	TRT_SH.CGS	A549	TRT_SH.CGS
KD_CYCLINS (*CCNL1*, *CCND1*, *CCNA1*, *CCNH*)	HCC515	TRT_SH.CGS	A549	TRT_OE
KEGG_CIRCADIAN_RHYTHM_MAMMAL	A549	TRT_SH.CGS	A549	TRT_SH.CGS
REACTOME_TRAF3_DEPENDENT_IRF_ACTIVATION_PATHWAY	A549	TRT_SH.CGS	A549	TRT_SH.CGS

**Table 6 ijms-23-00219-t006:** The identified significant (*p* < 0.05, connectivity score > 0.9) functional pathways from shRNA knock-down experiments in CMap using the up-regulated and down-regulated gene lists in Figure 6A as input.

src_set_id	Cell_Iname	Pert_Type	Genes
KD_APOLIPOPROTEINS	A549	TRT_SH.CGS	*APOB, APOC2, APOE*
KD_CYCLINS	A549	TRT_SH.CGS	*CCNL1, CCND1, CCNA1, CCNH*
	HCC515	TRT_SH.CGS	
KD_LYSINE_ACETYLTRANSFERASES	A549	TRT_SH.CGS	*KAT6B, KAT6A, NCOA3*
KD_NON_HOMOLOGOUS_END_JOINING	A549	TRT_SH.CGS	*RAD50, FEN1, XRCC4, XRCC5*
KD_V_TYPE_ATPASES	HCC515	TRT_SH.CGS	*ATP6V1A, ATP6V0C, ATP6V0B, ATP6V1F*
OE_NFKB_ACTIVATION	HCC515	TRT_SH.CGS	*CD40, FADD, LTBR, TNFRSF10A, TNFRSF10B, TNFRSF1A, BCL10*

## Data Availability

All data included in this analysis are publicly available with data access provided in the manuscript.

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
