# Peer review of "A Multi-Omics Network of a Seven-Gene Prognostic Signature for Non-Small Cell Lung Cancer"

_ijms, 2021, doi:10.3390/ijms23010219_

Round 1

Reviewer 1 Report

Dear authors,

It was my pleasure to read your research paper entitled” A multi-omics network of a 7-gene prognostic signature for 2 non-small cell lung cancer” which is presenting a multi-mics approach of a 7-gene prognostic signature for NSCLC.

Due to its high incidence, lung cancer represents a major issue in medical field and alternative recognition methods are needed, mostly for early detection of this disease. 

The scientific data presented in the manuscript is supporting the hypothesis and the results are well described. The methodology is good and the study design well structured. The number of analyzed samples is high enough to provide solid scientific data.

The selected references are adequate and sustain the results and discussion that are presented. 

I have one minor observation regarding a typing error

  • Row 220 (table 1) – please correct the word “epxression”- to “expression”
  • Row 224 (table 2) – please correct the word “epxression”- to “expression”

I recommend the publication of the manuscript!

Author Response

Thank you very much for your very positive review! We have corrected these typos in our revised manuscript.

Reviewer 2 Report

The manuscript presented by Guo and colleagues describes the genes' and proteins network related to a 7-gene prognostic signature in NSCL cancers. The manuscript is too long and I have got lost several times in reading. I suggest authors giving meaningful information in results without comments and shortening the manuscript in order to render it more readable.

The authors state in the abstract: "There is an unmet clinical need to identify patients with early-stage non-small cell lung 16 cancer (NSCLC) who are likely to develop recurrence and to predict clinical benefits of chemotherapy, with or without subsequent atezolizumab." What is the added information to the above mentioned sentence that the authors give with their study? 

Rows 178-179: " We acknowledge that many factors other than the response to radiotherapy may impact survival." Please, delete from result section.

Rows 181-184: "Down-regulation of SERPINB5 by small interfering RNA increased the resistance of NSCLC H460 cells to ionizing radiation [39], consistent with the observed association between SERPINB5 up-regulation and prolonged survival in TCGA NSCLC patients receiving radiotherapy."  Please, move to discussion.

Rows 195-206: "Cisplatin-etoposide combination has been successful in treating small cell lung cancer. Long-term daily administration of 196 oral etoposide in combination with cisplatin has been used to treat NSCLC [43], and cisplatin-etoposide has comparable efficacy as carboplatin-paclitaxel when used with con- 198 current radiotherapy for patients with stage III unresectable NSCLC [44]. Paclitaxel is a  tubulin-binding agent and is used in combination with a platinum-based compound for  NSCLC treatment [45]. Gefitinib and erlotinib are epidermal growth factor receptor  (EGFR) tyrosine kinase inhibitors for treating advanced NSCLC, with comparable effects on patient survival and overall response rate [46]. Docetaxel is an effective second-line treatment for NSCLC following platinum-based chemotherapy [47]. The combination of  pembrolizumab plus docetaxel was well tolerated and offered clinical benefits in ad vanced NSCLC patients after platinum-based chemotherapy, including patients with  EGFR variations [48]." Those are not results, please remove from this section.

Rows 226-239. It is not clear to me, the reason to correlate mRNA with protein levels. Please explain. Furthermore, in Fig 3C despite the p value, I cannot see a strong correlation between mRNA and protei level of CDH3 (r= 0,24!). 

Rows 383-387: "Low XRCC5 protein expression was more prevalent among squamous cell carcinoma compared with  lung adenocarcinoma compared with normal lung tissues [51]. Cell division cycle associated 3 gene (CDCA3) mRNA and protein expressions were increased in NSCLC compared with normal tissue, and high levels of CDCA3 was associated with poor prognosis [52]." If I have understood correctly, these are not your results. therefore remove from the results and discuss in the proper section of the manuscript.

Drugs repositioning as described in 4.8 and 4.10  is not clear to me. Please, try to explain better. Drugs repositioning is an important information, therefore it should be clearly presented.

Rows 60-65: "The protein expression of CD27 quantified with ELISA had a strong correlation with its mRNA expression in NSCLC tumors [10]. CD27 is a new generation of immune checkpoint inhibitor (ICI) [11]. There are promising results in phase I/II clinical trials of using CD27 as adjuvant immunotherapy for multiple tumor types [12,13]. CD27 agonist antibodies, either alone or in combination with anti-PD1, can improve the therapeutic efficacy of cancer vaccines 65 and immunotherapy in general [14,15]."  and rows 495-502 are highly similar "The protein expression of CD27 quantified with ELISA had a strong correlation with its mRNA expression  in NSCLC tumors [10]. CD27 is a new generation of immunotherapy target [11,71], currently being tested in phase I/II clinical trials for multiple tumor types with promising results [12,13]. CD27 agonist antibodies, either alone or in combination with PD1-blockade, can improve the efficacy of cancer vaccines and immunotherapy [14,15]. The synergy  between PD-1 blockade and CD27 stimulation for CD8+ T-cell driven anti-tumor immunity was reported [15], indicating the therapeutic potential of CD27 in neoadjuvant PD-1 blockade in resectable lung cancer [72]." Please, maintain this information in only one section of the article.

Rows 520-523 "The multi-omics network genes had a higher percentage (82.9%) of correlated mRNA and protein expression in CCLE NSCLC cell lines compared with the rest of the genome (66.3%). These results further substantiate the potential clinical implications of the multi-omics network of the 7-gene signature." Why? Please, explain

In the conclusion a paragraph describing the limits of the study would be appreciated.
